# Do general practitioners working in or alongside the emergency department improve clinical outcomes or experience? A mixed-methods study

Arabella Scantlebury ![ORCID],[1] Joy Adamson ![ORCID],[2] Chris Salisbury ![ORCID],[3] Heather Brant ![ORCID],[4] Helen Anderson,[2] Helen Baxter,[5] Karen Bloor ![ORCID],[2] Sean Cowlishaw,[3,6] Tim Doran,[2] James Gaughan ![ORCID],[2] Andy Gibson,[4] Nils Gutacker,[7] Heather Leggett ![ORCID],[2] Sarah Purdy,[5] Sarah Voss ![ORCID],[4] Jonathan Richard Benger[4]

**Correspondence to**
Professor Jonathan Richard Benger;
jonathan.benger@uwe.ac.uk

## ABSTRACT

**Objectives** To examine the effect of general practitioners (GPs) working in or alongside the emergency department (GPED) on patient outcomes and experience, and the associated impacts of implementation on the workforce.

**Design** Mixed-methods study: interviews with service leaders and NHS managers; in-depth case studies (n=10) and retrospective observational analysis of routinely collected national data. We used normalisation process theory to map our findings to the theory's four main constructs of coherence, cognitive participation, collective action and reflexive monitoring.

**Setting and participants** Data were collected from 64 EDs in England. Case site data included: non-participant observation of 142 clinical encounters; 467 semistructured interviews with policy-makers, service leaders, clinical staff, patients and carers. Retrospective observational analysis used routinely collected Hospital Episode Statistics alongside information on GPED service hours from 40 hospitals for which complete data were available.

**Results** There was disagreement at individual, stakeholder and organisational levels regarding the purpose and potential impact of GPED (coherence). Participants criticised policy development and implementation, and staff engagement was hindered by tensions between ED and GP staff (cognitive participation). Patient 'streaming' processes, staffing and resource constraints influenced whether GPED became embedded in routine practice. Concerns that GPED may increase ED attendance influenced staff views. Our quantitative analysis showed no detectable impact on attendance (collective action). Stakeholders disagreed whether GPED was successful, due to variations in GPED model, site-specific patient mix and governance arrangements. Following statistical adjustment for multiple testing, we found no impact on: ED reattendances within 7 days, patients discharged within 4 hours of arrival, patients leaving the ED without being seen; inpatient admissions; non-urgent ED attendances and 30-day mortality (reflexive monitoring).

**Conclusions** We found a high degree of variability between hospital sites, but no overall evidence that GPED increases the efficient operation of EDs or improves clinical outcomes, patient or staff experience.

**Trial registration number** ISCRTN5178022.

## STRENGTHS AND LIMITATIONS OF THIS STUDY

⇒ National evaluation of the impact of general practitioners working in or alongside emergency departments in England.
⇒ Mixed-methods approach using a large qualitative data set (413 interviews, 142 non-participant observation) and routine national data sets involving multiple stakeholders across 64 emergency departments gave us a service wide and detailed understanding of the impact of general practitioners working in or alongside the emergency department.
⇒ Our data apply to England only and so may not be generalisable to other countries and healthcare settings.
⇒ Our quantitative analysis was limited to routinely available data and so our analysis was dependent on key performance indicators and what is routinely collected and reported.

## INTRODUCTION

There were almost 24 million attendances at hospital emergency departments (EDs) in England in 2017–18, an increase of 22% since 2007/2008.[1] This continues a long-term trend of increasing demand for urgent care at EDs that has also been observed in many other countries.[2] Workload pressures within these departments can lead to adverse effects on the quality of patient care, patient safety, clinical outcomes, patient satisfaction and staff job satisfaction.[3] One important measure of the performance of EDs in England is the target that 95% of patients should be admitted, transferred or discharged within 4 hours of arrival. This target has not been

met nationally since 2015, with performance declining every year.[1]

About one-fifth of patients attending EDs could be managed by general practitioners (GPs) in primary care settings, although estimates of this proportion vary widely depending on the definitions used.[4] Research suggests that the reasons patients choose to attend an ED with problems suitable for general practice include: the perceived urgency of the situation, the belief that they need care only available in hospitals, the convenience of obtaining care at any time without an appointment, barriers to accessing general practice and a lack of awareness of available primary care services.[5–7]

Several different policy initiatives have been proposed to address rising ED demand, and to allow EDs to focus on patients with the most urgent need.[8–10] These responses fall into three main categories: a triage step before patients attend EDs, such as a telephone advice line or 'streaming' at the front door of the ED to direct patients to alternative services off-site; better provision of alternative services (such as nurse-led walk-in services and urgent treatment centres); improved access to GP services for people attending EDs. The latter approach can be achieved either by colocating GP services alongside EDs at hospital sites, or by employing GPs to work within EDs to see selected patients. It has been suggested that GPs in or alongside the ED have the potential to improve patient care, and to reduce waiting times, unnecessary investigations, hospital admission rates and costs,[11] but evidence to substantiate these claims is limited.[12–16] The introduction of these services was accelerated in 2017, when the UK government provided £100 million of capital funding to support hospitals in England to provide a GP working in or alongside the ED,[17–19] as part of a comprehensive plan to reduce the growth of lower acuity patients attending EDs.[10] The aim of our research was to examine the effect of GPs working in or alongside ED (GPED) on patient outcomes and experience and the associated workforce and system impact. To incorporate all aspects of the research—both evaluative and the experiences associated with implementation—we have situated our work in the normalisation process theory (NPT) framework.[20 21]

## METHODS
### Design
We completed a mixed-methods study including interviews with service leaders and NHS managers, in-depth case studies and a retrospective observational analysis of routinely collected national data. This approach enabled us to obtain a service-wide understanding of the impact of GPED on the urgent care system, the associated workforce and patient care.[22–25] Details of the study methodology have been published previously.[26]

### Theoretical approach
We drew on NPT, which has been widely used to understand how and why things do or do not become embedded

---

> **Box 1   The four core constructs of NPT, adapted for use in the GPED study**
>
> **Coherence:** Do staff understand why GPED has been implemented?
> **Cognitive participation:** Are staff engaged and committed to GPED, and what are the factors that promote and/or inhibit this commitment?
> **Collective action:** Are participants using GPED and what are the factors that promote and/or inhibit them from using GPED?
> **Reflexive monitoring:** Have staff appraised GPED and its impact on practice?
> GPED, general practitioners working in or alongside the emergency department; NPT, normalisation process theory.

into routine practice.[20] Through its four core constructs of coherence, cognitive participation, collective action and reflexive monitoring (box 1)[20 21] NPT can support both the understanding and evaluation of the implementation of organisational innovations such as GPED.[27] Its use has been supported by empirical studies using both qualitative and quantitative methods—therefore it was a particularly useful framework to apply in this context, given our study aims.[28 29]

NPT enabled us to integrate our qualitative and quantitative data; examining the extent to which GPED had become a part of routine practice and highlighting the related impact on patients and staff.

### Qualitative data collection and analysis
Qualitative data collection (table 1) consisted of non-participant observation of 142 individual clinical encounters and 467 semistructured interviews with key stakeholders (policy-makers, service leaders, ED staff, GP, patients and carers). Qualitative data were distributed across 64 NHS EDs in England, 10 of which were in-depth case study sites. Data collection explored the impact of GPED from the perspectives of key stakeholders as well as the policy's background and factors affecting implementation (see online supplemental files 1 and 2, eg, topic guides). Following initial familiarisation and independent coding, the qualitative team, through a series of roundtable discussions and workshops with our patient collaborators, developed a coding framework (online supplemental additional file 1). The coding framework, in conjunction with pen portraits of our ten case sites,[30] was used to facilitate cross-case comparisons and formed the basis of our main thematic analysis.[31] Initial analysis identified ten key themes (table 2)—these included contested policy, which reflected stakeholder views on the concept of GPED and structural implementation relating to site level responses to the introduction of GPED. In addition, we identified eight themes which were factors our participants predicted would be affected by GPED (at time 1 qualitative data collection): Performance against the 4-hour target; use of investigations; hospital admission; patient outcome and experience; service access; staff recruitment and retention; workforce behaviour and experience; resource use. We have collectively termed these eight themes as 'domains of influence',[32] which we

**Table 1** Qualitative data collection

| | Policy-makers | Service leaders | | Case sites | | |
|---|---|---|---|---|---|---|
| | Time 1 | Time 1 | Time 2 | Time 1 | Time 2 | Time 3 |
| Type of data collected | Semistructured telephone interviews | Semistructured telephone interviews | Semistructured telephone interviews | Semistructured face-to-face and telephone interviews, non-participant observations. | Semistructured face-to-face and telephone interviews | Semistructured face-to-face and telephone interviews, non-participant observations. |
| Aim of data collection | In-depth understanding of GPED policy and implementation from key informants | Broad perspective of GPED implementation and current provision from a range of EDs | Broad perspective of GPED implementation and current provision from a range of EDs | In-depth understanding from a small number of case sites | Brief 'check in' visits to assess any interim changes in GPED services | In-depth understanding from a small number of case sites |
| Period of data collection | December 2017 to January 2018 | August 2017–September 2018 | February 2018–February 2019 | November 2017–December 2018 | June–October 2018 | November 2018–December 2019 |
| No of EDs | Not applicable | 64 | 30 | 10 | 5 | 10 |
| Stakeholder groups and organisations represented | NHS England and Improvement, Department of Health and Social Care, Clinical Commissioning Groups, NHS Trusts, Royal College of Emergency Medicine, GPs | Chief executives, chief operating officers, lead clinical leads, nurses and ED managers | Chief executives, chief operating officers, clinical leads, lead nurses and ED managers | GPs, ED doctors (juniors, registrars, consultants), Nurses (streaming, triage, emergency nurse practitioner), patients and carers | GPs, ED doctors (juniors, registrars, consultants), Nurses (streaming, triage, emergency nurse practitioner), patients and carers | GPs, ED doctors (juniors, registrars, consultants), Nurses (streaming, triage, emergency nurse practitioner), patients and carers |
| Total number of participants | 10 policy-makers | 57 service leaders | 26 service leaders | 124 health professionals 94 patients/carers 83 non-participant observations. | 20 health professionals | 82 health professionals, 54 patients/carers, 59 non-participant observations. |

ED, emergency department; GP, general practitioner; GPED, general practitioner; GPED, general practitioners working in or alongside the emergency department.

**Table 2** Qualitative and quantitative data integration

| Theme | Qualitative | Quantitative |
|---|---|---|
| Contested policy | Qualitative interviews with policy-makers and service leaders, health professionals, patients and carers. Non-participant observation | |
| Performance against the 4 hours target | Qualitative interviews with policy-makers, health professionals, patients and carers. Non-participant observation | HES data: percentage of patients discharged within 4 hours of arrival |
| Use of investigations | Qualitative interviews with policy-makers, health professionals, patients and carers. Non-participant observation | |
| Hospital admissions | Qualitative interviews with policy-makers, health professionals, patients and carers. Non-participant observation | HES data: ED attendances that resulted in hospital admission |
| Patient outcome and experience | Qualitative interviews with policy-makers, health professionals, patients and carers. Non-participant observation | HES data: patients who left without being seen<br>HES data: Unplanned reattendance at the ED within 7 days<br>HES data: 30-day mortality |
| Service access | Qualitative interviews with policy-makers, health professionals, patients and carers. Non-participant observation | HES data: non-urgent (described previously as 'unnecessary') ED attendances<br>HES data: Volume of attendances |
| Staff (recruitment, retention) | Qualitative interviews with policy-makers, health professionals, patients and carers. Non-participant observation | |
| Workforce (behaviour, experience) | Qualitative interviews with policy-makers, health professionals, patients and carers. Non-participant observation | |
| Resource use | Qualitative interviews with policy-makers and service leaders, health professionals, patients and carers. Non-participant observation | |
| Structural implementation | Qualitative interviews with policy-makers, health professionals, patients and carers. Non-participant observation | |

ED, emergency department; HES, Hospital Episode Statistics.

have then used as outcome measures in our evaluation of GPED.

## Quantitative data and analysis

We completed a retrospective observational analysis of routinely collected Hospital Episode Statistics (HES) data between April 2018 and March 2019 from 40 English hospitals that were selected for their ability to provide complete data on the times of day when GPED services were available. Differences in GPED service availability between EDs at the same time of day were used to assign patients quasi-randomly to treatment or control groups at each hour of the day. Outcomes measured were: percentage of patients discharged within 4 hours of arrival; ED attendances that resulted in hospital admission; patients who left without being seen; unplanned reattendance at the ED within 7 days; 30 day mortality; non-urgent ED attendances (described previously as 'unnecessary' and identified using a defined methodology)[33]; volume of ED attendances. Each outcome was analysed separately using two-way fixed effects. Outcomes for patients attending different EDs at the same time of day were compared, exploiting variation in the timing of availability of GPED within the day at different EDs. Further details of this analysis have been published previously.[34] The potential net cost savings were explored using a comparative approach based on the results of this analysis.[35] We also conducted a survey of the GPED workforce at our 10 case sites, however, as these results did not materially alter our overall findings they are not reported here.[35]

## Mixed-methods analysis

In addition to individual quantitative and qualitative analyses, we conducted higher-level synthesis to integrate the study findings using a triangulation protocol that combined different methods to gain a more complete picture.[36] Quantitative findings were grouped under the qualitative themes described above (table 2). We then

**Table 3** Coherence—do stakeholders understand why GPED has been implemented

| Questions | Themes | Illustrative data |
|---|---|---|
| Does GPED have a clear purpose and did participants have a shared sense of its this purpose? Will GPED fit with the overall goals and activity of the organisation? Is it clearly distinct from other interventions? What benefits will the intervention bring and to whom? | ***Contested policy*** ► The implementation of GPED was considered rushed, and to be based on conflicting guidance. ► Some stakeholders had difficulty understanding how GPED differed from other previously unsuccessful attempts to introduce GPs into the ED. ► It was uncertain how GPED, or the associated capital funding initiative, differed from previous and existing interventions. ► Variations in local context, ED demand and existing GP services in the ED resulted in GPED being interpreted and implemented differently. ***Domains of influence*** ► GPED is difficult to describe, distinguish from other interventions and participants do not have a shared sense of its purpose. ► Stakeholders disagreed on the potential impacts of GPED, with positive, neutral or negative effects predicted for the majority of the eight identified domains of influence: (1) Performance against the 4 hour target; (2) Use of investigations; (3) Hospital admissions; (4) Patient outcome and experience; (5) Service access; (6) Staff recruitment and retention, (7) Workforce behaviour and experience; (8) Resource use. | *'I think it adds to the mix. I think that it was not a very well thought through policy decision … It was never part of the urgent, the care, the Keogh review of urgent emergency care to have GPs in ED. Now that review focused much more on NHS 111 and also trying to create consistency … So having GPs in ED, was outside of that policy strand. So, and it was dropped in a very, at very great speed and without a great deal of thought." (Interview with service leader)* *"You know, it isn't a sufficient evidence base to work from. You could have looked at the North East of England, I'm taking this call just now and said, you know, six of the top ten performers nationally sit in the North East, alright, and that tells us something about the system… and I think that, if we're going to use examples as a way of developing policy, that would have been a better way of looking at it." (Interview with Policymaker)* *"Whilst we started with a very clear - here's the Luton model, it became, obviously when trusts came to implement it locally that due to various circumstances that were very specific to their trust and their community, the Luton model just wasn't appropriate. So, I think what we've ended up with is a range of different models. So, you couldn't look at GP streaming and say what we've got in place now is the same in every trust in the country because there's almost certainly … there's huge variation in practice around how they're running.' (Interview with policymaker)* |

ED, emergency department; GPED, general practitioners working in or alongside the emergency department; GPs, general practitioners.

**Table 4** Cognitive participation—are people committed to using GPED and what are the factors that promote and/or inhibit this commitment

| Questions | Themes | Illustrative data |
|---|---|---|
| Did stakeholders see the point easily? Were stakeholders prepared to invest time, energy and work in GPED? | ***Contested policy*** ► There was doubt whether GPED, as a single initiative, could fix complex problems in the healthcare system. ► GPED policy development was criticised, as was the fact that it was based on limited evidence and patient and clinical consultation. This reduced stakeholders' commitment to ensuring it was embedded into routine practice. | *'Because it [GPED] is cheaper than re-investing in social care. Preventing inappropriate admissions is right, but it doesn't solve all the problems in primary care—those patients that do need to be seen and do need support in the community/social care, [GPED] is not a long-term solution.' (Interview with service leader)* *'It [streaming criteria] should be fixed, but, as I said, depending on who you speak to, it does waver slightly on what practitioners and GPs are willing to see. So, it's a bit of a grey area really. It depends who you're working with really. I don't … yes. So, it's not fixed. It should be really.' (Interview with Advanced Nurse Practitioner at Case site Redwood).* *'Actually, looking at X-rays and ECGs is, it becomes a bit of a, a dying art in General Practice, if you're not looking at those sorts of things on a daily basis, and what we provide again is allowing GPs the ability to keep those sort of clinical skills up and running, when I think that, and I think that's the attractiveness about doing this.' (Interview with Urgent Care Centre clinical lead at case site Teak)* |

GPED, general practitioners working in or alongside the emergency department.

mapped our study findings onto the four core constructs of NPT (tables 3–6).[20] Given the inter-related nature of the NPT constructs this process was undertaken by two researchers (JA and AS).

### Patient and public involvement

Patients and members of the public were involved throughout the development and delivery of this research. We formed a group of 10 public contributors with a wide variety of experiences of ED services. Throughout the study, members of the group were involved in regular workshops and meetings where they were asked to assist in interpreting the qualitative and quantitative data and support the development of our mixed-methods synthesis. For instance, our lay contributors highlighted the central role played by the streaming nurse in GPED, which led to a further analysis of qualitative data surrounding streaming that has been published previously.[37] Two members of the group were also full members of the Study Steering Committee.

### RESULTS

Tables 3–6 show how the themes from our qualitative and quantitative data map onto the four constructs of NPT.

### Coherence: do stakeholders have an understanding of why GPED was implemented?

For a health policy to be adopted into routine practice, there needs to be a shared sense of its purpose. Many stakeholders understood that GPED was being introduced as a direct response to rising pressures in EDs and as a potential mechanism for improving ED performance. Despite this, all stakeholder groups suggested that GPED was a rushed policy that lacked clear and consistent guidance. The fact that the policy was believed to originate largely from discussions between the Secretary of State for Health and NHS England, leading to 'top down' implementation, and the lack of evidence supporting the clinical and cost-effectiveness of GPED were further causes of concern.

The decision to introduce GPED nationally was also based on the perceived success of a GPED service that had been implemented at a single NHS site—Luton and Dunstable (L&D). The rationale for choosing L&D as the national exemplar over other high-performing EDs was unclear, particularly given that it was difficult to determine whether the perceived success of L&D was due to GPED or the simultaneous introduction of other initiatives within the organisation. Associated with this were concerns that GPED failed to acknowledge local context and variations in demand for ED services, varying patient populations and pre-existing or prior attempts to implement GPED services.

This led to stakeholders questioning the generalisability of the national policy, and as a result GPED was interpreted differently with a range of models implemented throughout the NHS in England.[38 39]

There was widespread disagreement at an individual, stakeholder and organisational level about the purpose and potential impact of GPED. Despite disagreeing about the 'direction of effect,' stakeholders agreed on the areas of the healthcare system and patient care that GPED was most likely to affect. We categorised these as eight themes as 'domains of influence' (table 3),[32] which were subsequently used as the outcomes for our evaluation of GPED.

### Cognitive participation: are people committed to using GPED and what are the factors that promote and/or inhibit this commitment?

The way in which GPED policy was designed and implemented, along with challenges in translating a national policy to meet local service and population needs, caused some to view GPED as a 'sticking plaster solution' to ED pressures. For many, the rise in ED attendances was driven by wider, more complex issues across health and social care, which were often deemed to be the result of deficiencies elsewhere in the system. As a result, there was doubt that a single initiative such as GPED could provide the solution. This lack of buy-in from stakeholders was reflected during interviews with service leaders and policymakers where alternative solutions for improving ED performance were proposed. For example, investment in social care and mental health services were considered to have a greater potential for impact.

Embedding GPED into existing practice requires commitment from key stakeholders. Emphasis was placed on the importance of streaming nurses and GPs working together to stream patients from ED to GPED. Despite many sites trying to ensure consistency through the development of streaming protocols, the challenges of disseminating and adhering to these protocols, reliance on locum and/or part-time GPs and frequent rotation of streaming nurses meant that the definition of a patient suitable for GPED varied between and within professional groups. This, combined with the cultural differences in how GPs and ED clinicians work, and their inherently different approaches to risk, was a source of tension that in some cases resulted in patients not being accepted by GPED and sent back to ED.

Whether GPED models gave GPs access to investigations such as X-rays and blood tests varied across case sites and reflected the different interpretations of the purpose of GPED and varying local contexts. Some individuals considered giving GPs access to investigations and diagnostic tests as crucial to the model's effectiveness by supporting GPs to treat a broader range of patients and refer to inpatient specialties. However, others felt that doing so asked GPs to work beyond their clinical competency—some staff felt that there was a shortage of GPs with the skills required to interpret some ED diagnostic tests, and an upskilling of the GP workforce would therefore be required. As a result, some GPs were asked to work as they would in general practice, while other services preferred those with prior ED experience.

**Table 5** Collective action—are people using GPED and what are the factors that promote and/or inhibit them from using GPED

| Questions | Themes | Illustrative data |
|---|---|---|
| What effect will GPED have on the ED and health service?<br><br>How will the intervention affect the work of patients and staff?<br><br>Will staff require further training?<br><br>What impact will it have on division of labour, resources, power and responsibility between different professional groups?<br><br>What are the factors that promote and/or inhibit them from using GPED? | **Service access**<br>► Despite reports that GPs have been working in the ED for some time, only a small number of patients reported using GPED previously and expected to be streamed to GPED.<br>► Staff were concerned that GPED may create 'easy access to a GP', encouraging people to attend.<br>► Staff were concerned that patients attended the ED "inappropriately", and considered poor health literacy to affect how patients use GPED.<br>► GPED and 'Urgent Care' were considered confusing to patients and made navigating services more challenging.<br>► Analysis of HES data identified no significant impact on: volume of ED attendances; number of non-urgent (described previously as 'unnecessary') attendances<br><br>**Staff recruitment and retention**<br>► Staffing issues posed a major threat to the successful implementation and adoption of GPED.<br>► Nursing shortages and a lack of experienced nurses made the staffing of streaming services challenging.<br>► Streaming may change the role of nurses and divert them away from core ED work, making GPED settings less attractive. The psychological and physical impact of streaming may negatively affect nurses' work and willingness to invest energy and time in GPED.<br>► GPED may draw GPs away from traditional General Practice. ED staff vacancies created issues in the recruitment of ED and GP staff.<br>► To overcome recruitment issues, GPED needs to be viewed as an attractive place to work.<br>► The training and educational benefits that junior doctors may receive from working alongside GPED models were considered valuable, and may make them more committed to ensuring GPED is embedded into routine practice.<br><br>**Use of investigations**<br>► There was a lack of consensus as to whether GPED models should give GPs access to diagnostic testing, reflecting differing interpretations of the purpose of GPED and varying local needs. This caused tension between GP and ED staff and may make staff less likely to invest their time and energy into GPED.<br><br>**Workforce behaviour and experience**<br>► Good communication, trust and confidence between streaming staff and GPs are pivotal to the effectiveness of GPED.<br>► Staff were concerned about patients who attend the ED with conditions that could be treated in general practice, but had different perceptions of what constitutes a 'GPED appropriate patient.'<br>► Tensions between GPs and staff responsible for streaming decisions were common and reflected different attitudes to risk as well as staff members (ED and GP) protecting their own working environment – staff streamed patients to GPED, or back to ED during busy periods, to ease their respective workloads.<br>► Streaming protocols were developed to try to standardise streaming decisions and GPED acceptance criteria, however these were not consistently disseminated or followed.<br><br>**Structural implementation**<br>► Several implementation issues also affected the extent to which staff were able to embed GPED into their routine practice including structural support within the site, ensuring integrated information technology systems between ED and GPED and influencing factors relating to the GP's role such as ensuring a positive working environment and giving GPs access to investigations, where appropriate. | 'What appeals to me is that I can do a bit of acute general medicine, trauma etc. and I'm trained in that but equally, I can also lapse into what was my comfort zone ... and that works really well whereas when I'm feeling a bit more sort of "right, come on, you know, I can get into resus and I can learn a new thing' and I really enjoy that.' (Interview with GP at case site Juniper).<br><br>'The GP feels that one of the problems with the model, is that there is a need for experienced triage nurses in order for it to work, but the department has a high turnover of nursing staff and has difficulty retaining staff. There are only a couple of appointed nurses who have the experience required.' (Interview with ED Consultant at case site Redwood).<br><br>'ED's frightened to send anything away, so everything comes in. So, I don't blame the public for attending if they can see a GP within three-hours, rather than having to wait six, seven-days or two-weeks for an appointment. But I just wonder if it's made a demand for it, because you get people coming back to see the GP again in ED'. (Interview with Nurse at case site Rowan)<br><br>'Patients are savvy as well, tell you what they think they want you to hear in order to get them into the service they want to be seen by.' (Interview with Nurse at case site Linden).<br><br>'I think it's down to, obviously, your training, but also how risk averse you are, and some people are very risk averse and will just have a much lower threshold for streaming people into ED and then also the Urgent Care Centre, rather than directing appropriately, you know, taking that risk.' (Interview with Paramedic at case site Chestnut).<br><br>'It's going on long enough to do and we really just didn't know what else to do. I literally can't drive. I'm having trouble getting out of the house. We could do it today and get here and try and figure out what was going on, rather than go to the GP, the GP say, "Do this, then come back," then almost probably end up in the hospital as it's going there anyway, to do the same things. That was the decision really.' (Interview with patient at case site Hawthorn). |

ED, emergency department; GPED, general practitioners working in or alongside the emergency department; GPs, general practitioners.

**Table 6** Reflexive monitoring—have people appraised GPED and its impact on practice

| Questions | Themes | Illustrative data |
|---|---|---|
| Will it be clear what effects the intervention has had? How are users likely to perceive the intervention once it has been in use for a while? Is it likely to be perceived as advantageous for patients or staff? | **Performance against the 4-hour target and hospital admissions** ▶ There was no significant impact on the proportion of patients meeting the 4-hour target, or on the number of attendances resulting in a hospital admission. ▶ Variations in site-specific patient mix, GPED models and whether patients streamed to GPED were included in ED reporting statistics, combined with other factors that influence ED performance, may have contributed to the apparently limited effects of GPED. **Resource use** ▶ Any possible cost savings due to reduced reattendances were much smaller than the cost of providing the service itself. **Patient outcome and experience** ▶ Most patients saw the value of GPs working in or alongside the ED as long as they received appropriate care. ▶ Staff felt that GPED may negatively affect patient flow. ▶ There was no significant impact on the following performance indicators in the HES analysis: left without being seen; 30-day mortality; reattendance to the same ED within 7 days. | *'Yeah, I think that's really important, I think given the way the hospital performs with the Government's four hour target, I think it's a source of pride for the hospital for the Chief Exec.' (Interview with ED Consultant at case site Linden).* *'I don't necessarily think it is a bad thing to have it, but it provides marginal gains, and those marginal gains are, happening at a very high capital cost and an ongoing staffing cost and looking at the NHS budget as a whole, I think it's a shocking waste of money.' (Interview with ED consultant at case site Juniper)* |

ED, emergency department; GPED, general practitioners working in or alongside the emergency department; GPs, general practitioners.

## Collective action: are people using GPED and what are the factors that promote and/or inhibit them from using GPED?

At the time that GPED was introduced, general practice in England was facing a significant workforce crisis. This posed a real challenge both in terms of ensuring that EDs were able to recruit GPs to work in GPED and ensuring that in doing so workforce shortages elsewhere in the system were not exacerbated. Site staff suggested that to facilitate the recruitment of GPs, emphasis should be placed on ensuring that GPED was considered an attractive place to work and on supporting GPs to work within the scope of their practice. However, whether GPED was viewed as a positive role depended on the individual GP. For instance, while GPED may be appealing to those who wish to expand their work beyond traditional general practice, the scope, acuity and shift-based working that are typical of the ED may contradict why many individuals chose to become a GP in the first place.

Ensuring that streaming is undertaken by experienced streaming nurses was also considered pivotal to an effective GPED service. However, nursing shortages, the psychological and physical burden of streaming on nurses and the potential for streaming to divert nurses away from their routine ED work meant that recruiting nurses to streaming roles was challenging.[37]

Our findings also identified several other factors that may promote or inhibit how staff use GPED, and the extent that it becomes embedded into routine practice (table 7). These were categorised as those relating to; workforce behaviour and experience (communication, trust and role-based cultural differences) and streaming and implementation issues (streaming protocols, interprofessional relationships and structural support).

Service leaders and site staff were concerned that giving patients 'easy access' to a GP, in a climate where general practice appointments may be difficult to obtain, could encourage patients to attend the ED rather than their own GP. Staff were particularly critical of patients for what they considered 'inappropriate ED attendance' (ie, attending the ED when they perceived alternative services would better meet their needs). While this was largely attributed to the potentially confusing range of services available, reorganisation and rebranding of existing services and perceived low levels of health literacy making service navigation difficult for patients, there were also some patients who were accused of deliberately 'playing the system'. For example, some patients were thought to deliberately bypass their GP and attend ED to access investigations, referrals or treatments. However, the reasons that patients chose to attend ED were complex, and in some cases, those that were considered by staff to have attended 'inappropriately' had been advised to attend the ED by other healthcare professionals and services such as NHS111, a pharmacy or their own GP.

However, our qualitative data provided numerous examples of situations in which experienced nurses were unable to determine whether a patient's complaint should be treated by general practice or the ED, suggesting that it may be unrealistic to expect patients to make the 'correct' choice on every occasion.

Despite these concerns among site staff, analysis of HES data found no association between non-urgent attendances and GPED or the absolute and relative volume of attendances and GPED.[34] Despite staff believing that GPED may encourage ED use, the qualitative data highlighted that patients attend the ED for a variety of reasons,

**Table 7** Success factors for the implementation of GPED

| Success factor | How can this be addressed? |
|---|---|
| Streaming | No single model for effective streaming was identified. The factors listed below should be considered when developing future streaming models. |
| The experience and seniority of streaming nurses | Effective streaming requires high levels of clinical knowledge, critical thinking, clinical decision-making and balancing clinical risks. Streaming should be undertaken by senior nurses. |
| The skills, confidence and abilities of GPs | Professional groups had different opinions as to what can be considered a 'GP appropriate' patient. To alleviate tension between staff there needs to be a shared understanding of streaming protocols and an awareness of the skills and scope of practice of GPs. Recruiting experienced and clinically knowledgeable GPs who are willing to adapt and see a broader range of patients is helpful. |
| Interprofessional relationships | Trust and confidence between professional groups is essential. Co-location does not automatically ensure collaboration. Individuals naturally work within professional norms. Effective communication and common goals mitigate tension. |
| Streaming protocols | Stakeholder clinicians (including streamers and GPs) should be involved in the development and regular review of protocols. These should be effectively communicated to all relevant practitioners. For streaming to be effective, streamers may need to deviate from protocols based on their clinical judgement. Staff should be supported to do this, while also considering strategies to mitigate against inappropriate deviation which may negatively impact patient care. |
| Streaming safety | Safety concerns limit the effectiveness of streaming strategies and sources of support are needed to ensure staff feel confident in their decision making. Clinicians should be involved in the development and regular review of protocols. These include effective pathways for managing deteriorating patients and returning streamed patients back to the ED when necessary. Consider ways to make the streaming process clearer for patients to navigate, to reduce repetition and patient frustration. Onward referrals were often a pinch point in the system, with patients at risk of increased waiting times or being overlooked. Guidance and support for streaming nurses experiencing complaints processes, litigation or professional registration issues should be provided. |
| Staffing | Less reliance on locum GPs and ensuring GPED shifts are covered consistently, and communicated effectively, promotes consistency. Recruitment of highly experienced and clinically knowledgeable GPs who are willing to adapt their practice to take on a broader range of work. Consider retention strategies to support current streaming nurses and to futureproof streaming by training and retaining adequate numbers of suitably experienced nurses. Streamers should be supported by their professional colleagues. Implement strategies to mitigate against burnout, prevent overload from additional responsibilities and positive promotion of streaming roles to make them attractive to nurses. |
| Leadership | Involve staff of all grades and from all key professional groups in the development and implementation of service planning, organisation and protocol development to counteract feelings of top-down change and encourage buy-in and support. |
| Physical environment | Consider the impact of the physical environment, for example, privacy at the streaming desk, safety of both staff and patients in isolated or exposed streaming areas, and for GPs located away from the ED and in off-site Hubs. Inadequate space can lead to overcrowding. Patients who have to queue more than can become confused and frustrated. Consider where GPs are placed to avoid feeling isolated and separated from the ED. |
| Integrated IT systems | Effective, easy to use and joined up information technology systems between ED, GPED and General Practice are essential for a safe working environment. |
| Structural support | Support for streamers should include specific training, regular supervision, audit and feedback. GPED models and streaming services should be planned and organised with involvement and buy-in from key stakeholders including streaming nurses and GPs. |

ED, emergency department; GP, general practitioner; GPED, general practitioners working in or alongside the emergency department.

and demonstrate reasoned decision-making in their service use. Only a small number of patients expected to see a GP, with the majority showing no awareness of GPED when interviewed. This is perhaps unsurprising given that sites often chose not to advertise GPED services to reduce the likelihood of driving an increase in ED attendances.

### Reflexive monitoring: have people appraised GPED and its impact on practice?

GPED is a complex intervention that has been introduced through a range of different models, into a complex and changing environment. EDs serve different patient populations and have different physical structures, staff mixes and care provision. In addition to this heterogeneity, the widespread uncertainty surrounding GPED operating hours and different governance arrangements across sites meant that there was variation in whether patients streamed to GPED were counted in nationally reported ED statistics. The challenges of using key performance indicators to evaluate national policies such as GPED was discussed by service leaders, who questioned their utility and described indicators such as the target that 95% of patients attending the ED should be admitted, transferred or discharged within 4 hours as 'blunt tools' for evaluating impact.

Our quantitative analysis showed no statistically significant improvement in a range of key performance indicators across several domains of influence including the '4-hour target', hospital admissions and patient outcomes and experience (patients leaving the ED without being seen and mortality at 30 days after an ED attendance). We did observe that GPED reduced the probability of unplanned reattendance within 7 days by 3.2% (OR 0.968, 95% CI 0.95 to 0.99), which equates to approximately 300 fewer reattendances per year for an average ED in England. After adjustment for multiple testing, however, this difference was no longer statistically significant, and was also not judged to be clinically significant. Possible cost savings associated with reduced reattendances (£30–37 000 per ED per year) were heavily outweighed by the cost of GPED services. In the hospitals for which we had data, the average length of time of operation of a GPED service was 11.1 hours per day. Assuming only one GP is present and including salary costs of the GP alone (potentially a substantial underestimate), this amounts to around £454 000 per ED per year. As a result, current GPED models do not appear to be an efficient use of healthcare resources.[35]

The majority of patients we interviewed valued GPED and considered it beneficial to have GPs in EDs. Patients were aware that GPED may relieve pressure on the ED, ensuring emergency doctors can deal with the 'real emergency cases' and were indifferent to the type of health professional that they saw as long as they received appropriate care. Similarly, the '4-hour target' was not a priority for patients, with many explaining that they were happy to wait longer as they understood that they were guaranteed to be seen and were waiting because priority was given to higher acuity patients. Despite this, staff raised concerns that GPED could negatively impact patient flow, as patients are required to disclose clinical information on multiple occasions before seeing a GP, which may create a backlog.

## DISCUSSION

The GPED study was commissioned to evaluate the impact of GPs working in or alongside EDs; a national policy implemented in response to rising pressures on EDs in England. GPED had no effect on a range of routinely collected ED performance measures. Despite considerable concern from health professionals that GPED may actually increase demand, we found no significant effect of GPED on ED attendances or reattendances within 7 days. This was supported by our qualitative analysis; most of the patients that we interviewed were unaware of GPED and had not changed their behaviour as a result. We observed confusion among patients, staff, service leaders and participating National Health Service (NHS) organisations as to the purpose of GPED, with a prevailing view that the main drivers of ED workload may be more related to an ageing population, high inpatient bed occupancy and a shortage of social care[40] than attendances by patients suitable for management in traditional general practice.

Early evaluations of GPED models of care in the UK and internationally suggested that placing GPs in the ED was a promising innovation.[41] Studies reported that GPED had the potential to reduce resource use,[42 43] and increase patient satisfaction.[44] Carson et al[45] found that the proportion of cases seen by GPs varied and that clinical and operational governance was often disjointed. In a survey of patients, Bickerton et al[46] found that while GPED offered patients a greater range of service provision, it also increased the risk of duplication and repeat attendance. More recently, in a relatively small study, Uthman et al found that GPs who saw patients in the ED used fewer resources without increasing reattendance and referred more patients to follow-up services.[47] In addition, service users appreciated simplified healthcare provision from a single point of access.[48]

It is not uncommon for early reports of new initiatives to be positive, but contradicted subsequently,[49] and our study is the largest of GPED services published to date. A similar phenomenon was observed previously in relation to nurse-led walk-in centres colocated with the ED, whereby initially positive reports were challenged by a subsequent large-scale evaluation that found 'no evidence of any effect on attendance rates, process, costs or outcome of care'.[50] Furthermore, our data demonstrate considerable heterogeneity, with the implication that while our overall result is null, GPED may still have beneficial effects in some locations and under certain circumstances. Our findings suggest that GPED implementation is highly sensitive to local context, and these contexts will govern the success of any particular scheme. This is consistent

with other evaluations of urgent and emergency care initiatives,[51] Investment in GPED appears justified only when the factors associated with success are in place (see table 7), and there is clear evidence of benefit at a local level. Where this evidence of local benefit is absent alternatives to GPED should be considered, such as improving provision and access in traditional general practice, both in and out of hours.

Our quantitative analysis used routinely available data, and it would be surprising if some of these measures (eg, 30-day mortality) were influenced by GPED. It has also been noted that patients eligible for GPED are often quick and easy to manage, do not breach the '4-hour target', are less likely to be admitted and do not contribute to crowding.[45 52] A recent realist review concluded that, despite GPs in ED being associated with a reduction in process time for non-urgent patients, this does not necessarily increase capacity to care for the sickest patients.[12] The main cause of ED crowding is perceived to be congestion in the flow of sicker patients into the hospital and a lack of beds, rather than absolute attendance numbers.[53]

The GPED study shows that even when a policy is mandatory and supported by dedicated capital funding, this does not guarantee successful or uniform adoption. Our findings highlight the complexities of translating policy into practice, and the importance of considering the extent that a government-led policy can be delivered at a local level. Previous evidence suggests that a common response to national policy is local adaptation, which can in turn lead to the implementation of different innovations to those that are originally proposed.[22] We found evidence of this, as interviewees often described a range of approaches to GPED that sometimes opposed the high-level policy messages that accompanied the provision of capital funding. It also remains uncertain whether revenue funding, as well as (or instead of) capital funding would have alleviated some of the noted challenges.

Our qualitative data also identified a range of factors that can facilitate implementation. We present these as a series of 'success factors' which may inform how services choose to implement future GPED models; or adapt existing ones (table 7). At several of our case study sites, these fundamentals had been overlooked and the result was a less coherent GPED service. However, it is important to note that even if all these 'success factors' are implemented, our findings do not present evidence that the resulting GPED service would have a positive impact on ED performance indicators or be cost-effective.

GPED is a new policy initiative, which has been evaluated by two large NIHR commissioned research studies (HS&DR Projects 15/145/04 and 15/145/06).[26 35 39 54] Further research evaluating its impact is therefore not recommended until the policy has been given time to embed into routine practice. Instead, priority should be given to evaluating existing performance measures and developing new, rapid methods to inform the development, implementation and evaluation of similar health policy initiatives (box 2).

---

> **Box 2  Implications for future research**
>
> 1. The utility and completeness of national routine data sets limit the ability to evaluate the impact of complex health initiatives across a range of outcomes. Patients and clinicians should be consulted to ensure that measures of 'success' include outcomes that are important to all stakeholder groups and consider how these can be captured.
> 2. The relationship and interface between general practice and secondary care is crucial to the future delivery of urgent and emergency care. Research to explore this relationship and different approaches to risk will inform future models of service development and delivery in the context of rising healthcare demand.
> 3. We identified particular ambiguity and uncertainty in relation to streaming in the ED. Further research to clarify the optimal approach to streaming in terms of patient outcome, safety and experience, and the wider implications of streaming on staff experience, is warranted.

### Strengths and limitations

We adopted a mixed-methods approach which consisted of 'big qualitative' data collection (467 interviews and 142 individual observations of clinical encounters) and quantitative analysis of national data sets to explore the impact of GPED. This approach, and the decision to interpret our study findings using NPT, provided us with an in-depth understanding of the impact of GPED. This highlighted the complex interplay of political, workforce and social factors that affect successful adoption of a health policy into routine practice.

Our data apply to England only, and so may not be generalisable to other countries and healthcare settings. In our quantitative analysis, it was not possible to identify from available data which staff members assessed and treated individual patients, so we could not separate patients treated by GPs from those treated by other ED staff to directly compare GP services to traditional models of care. We relied primarily on measures of general ED performance, such as attendances, patient flow and waiting times. We were also limited in our ability to collect data from the general practice and urgent care systems surrounding our case study sites, which significantly limited our ability to evaluate quantitatively the effect of GPED on the wider healthcare system. Our qualitative case study sites were selected purposively to be as representative as possible. However, participation by sites, and from staff and patients during data collection, was voluntary and so is unlikely to be exhaustive.

### CONCLUSION

Implementation of GPs working in or alongside the ED was highly subject to local context and microlevel influences. However, we found no consistent evidence of improvements in patient outcome or experience. This is summed up by our public contributors, who following presentation of the final study findings concluded:

GPED is not effective and should only be used where specific circumstances indicate that it may play a positive role.

**Author affiliations**
[1]York Trials Unit, University of York Department of Health Sciences, York, UK
[2]Department of Health Sciences, University of York, York, UK
[3]Centre for Academic Primary Care, School of Social and Community Medicine, University of Bristol, Bristol, UK
[4]School of Health and Social Wellbeing, College of Health, Science and Society, University of the West of England, Bristol, UK
[5]School of Social and Community Medicine, University of Bristol, Bristol, UK
[6]Department of Psychiatry, The University of Melbourne, Melbourne, Victoria, Australia
[7]Centre for Health Economics, University of York, York, UK

**Acknowledgements** The authors would like to thank the participants for their involvement in the qualitative study, and the patient and public contributors.

**Contributors** JRB, JA, HBa, KB, SC, TD, AG, NG, SP, CS and SV had the initial research idea and obtained funding for this study. Qualitative data collection and analysis were undertaken by HA, JA, HL, AS. Quantitative data collection and analysis were undertaken by KB, TD, JG, NG, HBr, SC. Mixed-methods analysis was undertaken by JA and AS. AS and JA drafted the paper and are joint first authors for the manuscript. JRB and CS assisted in drafting the manuscript. AS, JA, CS, HBr, HA, HBa, KB, SC, TD, JG, AG, NG, HL, SP, SV and JB critically reviewed, revised and approved the final manuscript. JRB is the study guarantor. The guarantor (JRB) accepts full responsibility for the work and the conduct of the study, had access to the data and controlled the decision to publish. The corresponding author attests that all listed authors meet the authorship criteria and that no others meeting the criteria have been omitted.

**Funding** Department of Health. National Institute for Health Research. Health Services and Delivery Research Programme. Grant number 15/145/06.

**Disclaimer** The views expressed are those of the authors and not necessarily those of the NIHR or the Department of Health and Social Care.

**Competing interests** None declared.

**Patient and public involvement** Patients and/or the public were involved in the design, or conduct, or reporting, or dissemination plans of this research. Refer to the Methods section for further details.

**Patient consent for publication** Not applicable.

**Ethics approval** Approval was obtained from East Midlands – Leicester South Research Ethics Committee (ref: 17/EM/0312); University of Newcastle Ethics Committee (Ref: 14348/2016) and the Health Research Authority (IRAS: 230848 and 218038). All participants provided informed consent before taking part in the qualitative study.

**Provenance and peer review** Not commissioned; externally peer reviewed.

**Data availability statement** Data may be obtained from a third party and are not publicly available. The deidentified patient-level data used for the quantitative component of this study, including information on mortality, were released by the data holders (NHS Digital, Office for National Statistics) under specific data sharing agreements and only for the purpose of this study. The data sharing agreements do not permit further sharing or publication of the data. Interested parties may seek to obtain data directly from the relevant data holders. Hospital Episode Statistics (HES) data are copyright 2018–2019, reused with the permission of NHS Digital through Data Sharing Agreement NIC-84254-J2G1Q. The data about the hours a general practitioner services was operating in emergency departments was collected by the authors specifically for this project. The authors are not able to place the original data into the public domain. The qualitative data we have acquired will not be available as our ethical approval does not permit the sharing of the entire data set.

**ORCID iDs**
Arabella Scantlebury http://orcid.org/0000-0003-3518-2740
Joy Adamson http://orcid.org/0000-0002-9860-0850
Chris Salisbury http://orcid.org/0000-0002-4378-3960
Heather Brant http://orcid.org/0000-0001-9608-7451
Karen Bloor http://orcid.org/0000-0003-4852-9854
James Gaughan http://orcid.org/0000-0002-8409-140X
Heather Leggett http://orcid.org/0000-0001-8708-9842
Sarah Voss http://orcid.org/0000-0001-5044-5145

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
