## [Reviewer comments · BMJ Open]

ARTICLE DETAILS

TITLE (PROVISIONAL)	Do General Practitioners Working in or Alongside the Emergency Department Improve Clinical Outcomes or Experience? A mixed methods study
AUTHORS	Scantlebury, Arabella ; Adamson, Joy; Salisbury, Chris; Brant, Heather; Anderson, Helen; Baxter, Helen; Bloor, Karen; Cowlshaw, Sean; Doran, Tim; Gaughan, James; Gibson, Andy; Gutacker, Nils; Leggett, Heather; Purdy, Sarah; Voss, Sarah; Benger, Jonathan

VERSION 1 – REVIEW

REVIEWER	Porter, Alison University of Swansea, College of Medicine
REVIEW RETURNED	29-Apr-2022

GENERAL COMMENTS	There is an interesting paper here about implementation presented under a title and with an objective that don't fit. The title and objective set this up for an evaluation paper - telling us what difference GPED make in terms of measurable outcomes. Instead, the authors tell as about the challenges and reality of introducing GPED models - interesting and useful, but the answer to a different question. The use of NPT is appropriate and generally handled well, and the qualitative data set is impressively large. I suggest that the authors be bold about presenting the paper as being a theoretically informed analysis of implementation. The abstract presents a mix of results, with the emphasis on implementation, though the main body of the paper is all about implementation. p5 line39/40 considers impact on attendance as part of the 'collective action' component of NPT - this is hard to grasp, as 'collective action' is normally associated with the work required to bring an innovation into use. p6 line 60 'streaming to alternative services' - are the authors regarding this as not an example of GPED? bearing in mind that those alternatives are often on-site p7 Line 14 - mentions capital funding to support introduction of GPED - it's worth considering whether capital funding was what was needed, as opposed to secure revenue funding p8 quali data collection - it's not clear how the dataset from 64 EDs relates to the 10 in depth case studies. Were different questions being addressed? where there separate processes of analysis? p8 liine 50 onwards - the presentation of themes is quite confusing, with 8 of the ten in a group of their own. Were the 8 'domains of influence' then used as outcome measures in the evaluation? things become a bit clearer in results Table 3, but could do with more clarity here. Table 1 for case study sites, Time 1 seems to overlap with Time 3 (they share a finish date). Why were there 3 time periods? were the
--

	authors hoping to observe change over time? Clarify data sources - routine data comes from 40 sites, which presumably were a sub-set of the 64. Were the chosen simply on the grounds of data availability? Page 12 quanti analysis. What is being compared? did all 40 EDs have GPEDs available for some hours in the day but not others? is the comparison between what happened when GPED operating (treatment) and not (control)? please clarify p12 worth setting up that the mapping of findings onto the NPT constructs happens in Table 3 (i was looking for it in Table 2) p14 reports some findings from patients (how many?). Worth discussing how much of collective action is work undertaken by patients (making choices about what to do). p23 Table of 'success factors' - are these about successful implementation, or success in terms of impact? p26 Implication 1 - relates to datasets. Can you justify this from evidence presented in this paper (as opposed to study findings reported elsewhere)?
--	--

REVIEWER	Heald, Adrian Salford Royal Hospitals NHS Trust, Endocrinology and Diabetes
REVIEW RETURNED	17-May-2022

GENERAL COMMENTS	Overall comments This very relevant paper provides evidence for an alignment of services that has been usual practice in a number of hospitals over the last 20 years and more, but has not until recently been analysed in terms of patient outcomes in the UK. It is clear that a great deal of work has gone into this paper. Results It would be good if the Results section could start with a narrative of the results as the first paragraph rather than going straight into a table. The results section is quite long and could be shortened. After a great start the paper moves into a complex, long and not hugely structured results section. Perhaps some the results could be summarised in the tables. Discussion The discussion would benefit form 1-2 more paragraphs to place this work in context in relation to international analysis in this area. The discussion is disproportionately short vs the results section. The 'null' result is somewhat surprising and more discussion of this would be helpful with policy suggestions in relation to what we can do better from here to spend the money wisely. I was witness to the early days of this approach ie embedding of general practioners in A/E Depts in 2005 and saw it working well in the NE of England. More explanation of why no overall benefit was seen would be helpful. I note the comment made on P17 of the paper (lines 42-49) in relation to the continuing success of this approach in the NE of England.
---

VERSION 1 – AUTHOR RESPONSE

Reviewer 1	
There is an interesting paper here about implementation presented under a title and with an objective that don't fit. The title and objective set this up for an evaluation paper - telling us what difference GPED make in terms of measurable outcomes. Instead, the authors tell as about the challenges and reality of introducing GPED models - interesting and useful, but the answer to a different question. The use of NPT is appropriate and generally handled well, and the qualitative data set is impressively large. I suggest that the authors be bold about presenting the paper as being a theoretically informed analysis of implementation.	We thank the reviewer for their positive comments and for highlighting the issue of implementation vs evaluation. Of course, this distinction is not straightforward – as proponents of NPT would advocate that “NPT provides a set of conceptual tools that support understanding and evaluation of the adoption, implementation, and sustainment of socio-technical and organisational innovations” (May et al. 2022). Therefore, using the implementation framework can be seen to include evaluative elements, and our paper does indeed do so with findings from the analysis of HES data incorporated in the relevant sections of the NPT framework, in addition to qualitative evaluative outcomes used throughout. Therefore, we are not in total agreement with the reviewer comment that ‘this is a different question’, they go hand in hand when this implementation framework is used. The aim of this paper is to present an overview of our findings in keeping with the published protocol (Morton et al., 2018, as referenced in the manuscript). To cover the full breadth of the work - and complementary methodologies - we have used NPT as a framework to provide a theoretical overview and an appropriate structure that allows us to combine our findings. However, the reviewer’s comments do highlight that this approach may have become ‘lost’ in the paper, and these aspects need to be pulled out more clearly. In response we have changed the wording to incorporate evaluation beneath the umbrella of implementation. We have also added a sentence at the end of the introduction and altered the methods section on NPT accordingly. May, C.R., Albers, B., Bracher, M. et al. Translational framework for implementation evaluation and research: a normalisation process theory coding manual for qualitative research and instrument development. Implementation Sci 17, 19 (2022). https://doi.org/10.1186/s13012-022-01191-x
The abstract presents a mix of results, with the emphasis on implementation, though the main body of the paper is all about implementation.	See above point. The objectives have been slightly reworded.
p5 line39/40 considers impact on attendance as part of the 'collective action' component of	It has been noted that the concepts of NPT are sometimes overlapping, and our

NPT - this is hard to grasp, as 'collective action' is normally associated with the work required to bring an innovation into use.	team debated how best to map our findings onto the NPT concepts. The reviewer's point reflects this uncertainty, and some of our own discussions. However, we felt it was most appropriate to bring in the quantitative outcome relating to attendance here, as it was under this heading that the 'work required to bring GPED into use' (including streaming and the use of investigations by GPs) was discussed. As a result of this work, staff were concerned about the impact on attendance, and we therefore felt it was most appropriate to note that these worries were unfounded at this point. Our view is that the flow of the paper would be interrupted if the point were highlighted elsewhere.
p6 line 60 'streaming to alternative services' - are the authors regarding this as not an example of GPED? bearing in mind that those alternatives are often on-site	To avoid confusion, we have changed the description slightly to include the word 'off-site'.
p7 Line 14 - mentions capital funding to support introduction of GPED - it's worth considering whether capital funding was what was needed, as opposed to secure revenue funding	This is a really important point, however, in the introduction we describe what actually happened to provide the necessary context. Instead, we have inserted a comment to reflect this point in the discussion.
p8 quali data collection - it's not clear how the dataset from 64 EDs relates to the 10 in depth case studies. Were different questions being addressed? where there separate processes of analysis?	It was 10 out of the 64 – the manuscript has been updated accordingly.
p8 liine 50 onwards - the presentation of themes is quite confusing, with 8 of the ten in a group of their own. Were the 8 'domains of influence' then used as outcome measures in the evaluation? Things become a bit clearer in results Table 3, but could do with more clarity here.	Yes, we agree this was confusing! We have reworded the text to make it clearer.
Table 1 for case study sites, Time 1 seems to overlap with Time 3 (they share a finish date). Why were there 3 time periods? were the authors hoping to observe change over time?	Thank you for identifying this error in Table 1, which we have corrected. We completed sequential visits to identify changes over time in our case study sites, particularly where new initiatives were being introduced.
Clarify data sources - routine data comes from 40 sites, which presumably were a subset of the 64. Were the chosen simply on the grounds of data availability?	Yes, this is the case and the paper states: "from the 40 English hospitals which were able to provide complete data on times of day when GPED services were available". We have adjusted the wording slightly to clarify this to make sure this detail is not lost.
Page 12 quanti analysis. What is being compared? did all 40 EDs have GPEDs available for some hours in the day but not others? is the comparison between what happened when GPED operating (treatment) and not (control)? please clarify	Yes, all 40 EDs had GPED available for some hours of the day, but not others. We have attempted to provide a high-level summary of the quantitative analysis, and reference another open access paper that describes this in full. We have considered other ways of describing the 'treatment' and 'control' groups, and the fact that differences in GPED availability between sites was used as the

	basis of this analysis, and have made some minor edits to improve the clarity of the text as a result.
p12 worth setting up that the mapping of findings onto the NPT constructs happens in Table 3 (I was looking for it in Table 2)	This was a 2-stage process. Initially the quantitative outcomes were mapped onto the qualitatively derived themes (shown in Table 2). Then these themes were mapped onto the NPT constructs (now in new tables 3-6). We have made this clearer in the text.
p14 reports some findings from patients (how many?). Worth discussing how much of collective action is work undertaken by patients (making choices about what to do).	The number of interviews with patients/carers is shown in Table 1, which summarises qualitative data collection. We agree the decision-making of patients is important and under collective action we have made the observation that the majority of patients arrived at the ED with limited, if any, knowledge of GPED. We have elaborated on this point, and included a quote from one of our patient participants. Given that we interviewed 148 patients, we do have a lot of relevant data, and this will be the focus of a separate publication.
p23 Table of 'success factors' - are these about successful implementation, or success in terms of impact?	These relate to implementation – and this is noted in the text, also making the distinction between implementation and impact. To improve clarity here we have changed the title of Table 4.
p26 Implication 1 - relates to datasets. Can you justify this from evidence presented in this paper (as opposed to study findings reported elsewhere)?	We have altered the wording here so this statement better reflects the evidence we have available.
Reviewer 2	
This very relevant paper provides evidence for an alignment of services that has been usual practice in a number of hospitals over the last 20 years and more, but has not until recently been analysed in terms of patient outcomes in the UK. It is clear that a great deal of work has gone into this paper.	We thank the reviewer for these positive comments.
Results It would be good if the Results section could start with a narrative of the results as the first paragraph rather than going straight into a table.	We have restructured the start of the results section accordingly.
The results section is quite long and could be shortened. After a great start the paper moves into a complex, long and not hugely structured results section. Perhaps some of the results could be summarised in the tables.	As suggested, we have moved some of the findings into summary tables. The overall NPT analysis table (originally Table 3) has been replaced with four smaller tables; one for each of the constructs of NPT, and with relevant data included (and removed from the main text).
The discussion would benefit from 1-2 more paragraphs to place this work in context in relation to international analysis in this area. The discussion is disproportionately short vs the results section.	We have expanded the discussion with additional paragraphs drawing on the previous UK and international literature.
The 'null' result is somewhat surprising	We have added text to explore and

and more discussion of this would be helpful with policy suggestions in relation to what we can do better from here to spend the money wisely.	explain the “null” result in more detail, with additional policy suggestions as suggested.
I was witness to the early days of this approach ie embedding of general practioners in A/E Depts in 2005 and saw it working well in the NE of England. More explanation of why no overall benefit was seen would be helpful. I note the comment made on P17 of the paper (lines 42-49) in relation to the continuing success of this approach in the NE of England.	As above, we have added text to explore the heterogeneity of our results and the potential reasons why an overall benefit was not observed. (Of note, the comment on page 17 relates to performance against the four hour target in the NE of England, rather than GPED services specifically).

VERSION 2 – REVIEW

REVIEWER	Heald, Adrian Salford Royal Hospitals NHS Trust, Endocrinology and Diabetes
REVIEW RETURNED	14-Aug-2022
GENERAL COMMENTS	This paper is very important in relation to NHS resource allocation and organisation of emergency care globally and I look forward to seeing it published.